

# The arctic seasonal cycle of total column $CO_2$ and $CH_4$ from ground-based solar and lunar FTIR absorption spectrometry

Matthias Buschmann[1], Nicholas M. Deutscher[1,2], Mathias Palm[1], Thorsten Warneke[1], Christine Weinzierl[1], and Justus Notholt[1]

[1]Institute of Environmental Physics, University of Bremen, Bremen, Germany
[2]School of Chemistry, University of Wollongong, Wollongong, NSW, Australia

*Correspondence to:* Matthias Buschmann (m_buschmann@iup.physik.uni-bremen.de)

**Abstract.** Solar absorption spectroscopy in the near infrared has been performed in Ny-Ålesund (78.9°N, 11.9°E) since 2002; however, due to the high latitude of the site, the sun is below the horizon from October to March (Polar Night) and no solar absorption measurements are possible. Here we present a novel method of retrieving the total column dry-air mole fractions (DMF) of $CO_2$ and $CH_4$ using the moon as a light source in winter. Measurements have been taken during the Polar Nights

from 2012 to 2016 and are validated with TCCON (Total Carbon Column Observing Network) measurements by parallel solar and lunar absorption measurements during spring and autumn. The complete seasonal cycle of the DMFs of $CO_2$ and $CH_4$ is presented and a precision of up to 0.5 % is achieved. Additionally a model comparison has been performed with data from various reanalysis models.

## 1 Introduction

Since 1992 a Fourier-Transform Infrared (FTIR) Spectrometer in Ny-Ålesund (78.9°N, 11.9°E) has been used for the ground-based observation of total column trace gas abundances in the Arctic via solar absorption spectroscopy (Notholt and Schrems, 1994). The measurements are taken within the Infrared Working Group (IRWG) of the Network for the Detection of Atmospheric Composition Change (NDACC). Since 2002, measurements in the near infrared (NIR) spectral region have been performed to retrieve the dry-air mole fractions (DMFs) of $CO_2$ and $CH_4$ (denoted here as $xCO_2$ and $xCH_4$) and other gases

(Warneke et al., 2005, 2006). These are, since 2005, part of the Total Carbon Column Observing Network (TCCON). Today, these measurements are widely used as validation for satellite products, in model comparisons and studies of sources and sinks.

A large limitation of the availability of these measurements is the absence of sunlight in the polar winter. At Ny-Ålesund, between October and March, the Sun is permanently below the horizon. However, during this period the moon is permanently above the horizon around full moon.

Moonlight has already successfully been used as a light source in retrievals of various trace gas concentrations via the FTIR spectrometer (FTS) in Ny-Ålesund in the middle infrared spectral region (Notholt et al., 1993, 1997; Notholt and Lehmann, 2003; Palm et al., 2010) and in Antarctica (Wood et al., 2004). Here the employment of liquid nitrogen cooled InSb and MCT detectors ensures low instrumental noise, even under low light conditions. In the NIR, i.e. $> 5000 \, cm^{-1}$, typically extended





range InGaAs diodes are used. Recently Fu et al. (2014) and Wong et al. (2015) showed the application of a thermo-electrically cooled InGaAs detector for the measurement of reflected sunlight spectra from the Los Angeles basin on a mountaintop site. The thermo-electrical (TE) cooling reduces the detector noise and allows for higher signal-to-noise ratios in the measured spectrum.

After initial tests at the Bremen TCCON site (Buschmann et al., 2015), a TE cooled InGaAs diode detector was implemented in the Ny-Ålesund FTS and a time series of $xCO_2$ and $xCH_4$, the total column dry air mole fraction, was obtained from spectra measured during polar night between 2012 and 2016. The resulting product is compared to TCCON solar measurements as well as model simulations from the MACC reanalysis model for $CO_2$ (v14r2 MACCCO2 (2016)) and for $CH_4$ (v10 MACCCH4 (2016)), the Jena $CO_2$ inversion CarboScope s04_v3.7 (JenaCO2, 2005) and the Carbontracker 2015 model (CT2015, 2016).

Together with the summer TCCON data from Ny-Ålesund, for the first time the whole seasonal cycle of $xCO_2$ and $xCH_4$ is presented.

    In Sections 2 and 3, this paper describes the measurement setup and the methods used to retrieve the dry air mole fractions. Section 4 describes the newly obtained time series and the comparison to TCCON. Finally we compare our results with model data in Section 5.

## 15   2   Setup

### 2.1   Measurement site

The instrument, a Bruker IFS 120-5HR, is located at the AWIPEV research station in Ny-Ålesund (78.92°N, 11.92°E). Measurements are taken under cloud-free conditions for both the NDACC and TCCON networks during summer, and lunar absorption and atmospheric emission measurements are performed in winter. In 2014/2015 the measurement setup was gradually

changed to a semi-automated system. The new system is able to automatically start a set of measurements without the need of an operator, which considerably increased the number of measured spectra. The performance of the instrument is monitored by reference cell measurements on a monthly basis and it is ensured that the phase error is smaller than $\pm 0.04$ and the modulation efficiency is $\pm 2\%$ of 1.0 up to a maximum optical path difference of $180\,\mathrm{cm}$.

### 2.2   Thermoelectrically cooled InGaAs diode

The sensitivity of the extended InGaAs diode used as a detector in standard TCCON near-infrared measurements is too small to obtain a sufficient signal-to-noise ratio from lunar irradiance. The introduction of a two-stage Peltier element cooling system attached to the back of the diode can reduce the dark current noise and thereby minimise overall detector noise. Generally the extension of the band-gap in the diode production process reduces the quantum efficiency. Therefore, a non-extended diode improves the signal-to-noise ratio; however, cooling the InGaAs diode affects its crystal structure and therefore widens the

band-gap, which leads to a shift of the diode's sensitivity range. The commercially available diode used here has a cut-on





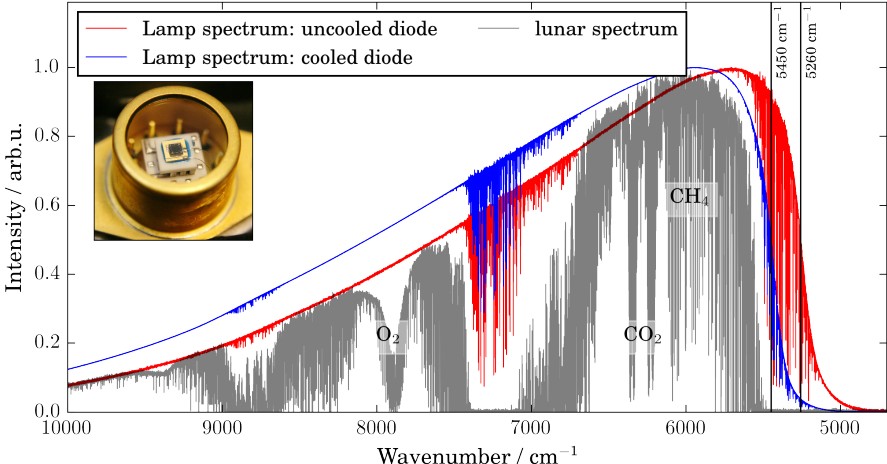

**Figure 1.** Example measurements of the InGaAs diode: Cooled (blue) and uncooled (red) lamp spectra. Note the indication of the cut-on wavenumbers. An averaged lunar spectrum is shown in gray and a picture of the diode was added.

frequency of about $5260\ \mathrm{cm}^{-1}$ in the uncooled and about $5450\ \mathrm{cm}^{-1}$ in the cooled state. The shift in sensitivity, an example of an averaged lunar spectrum, and a picture of the diode are shown in Fig. 1.

### 2.3 Availability of moonlight

The total number of potential lunar measurement hours can be calculated by excluding all times where the lunar elevation
is below the terrain height. Additionally, lunar phases with insufficient illumination (lunar phase $< 85\%$) and times where the solar zenith angle is smaller than $95°$ have to be excluded. Depending on lunar orbital parameters, the maximum number of measurement hours ranges from about $886\ \mathrm{h}$ in 2012 to $634\ \mathrm{h}$ in 2016. This is much less than the potential yearly solar measurement time of $3883\ \mathrm{h}$. The minimum lunar zenith angle is $57.13°$ (2012) and $60.84°$ (2016) compared to a minimum solar zenith angle of $55.47°$. The actual possible time available for near-infrared measurements, of course, further depends
on clear sky conditions and other scheduled FTS experiments. The number of measurements was increased by switching to a semi-automated measurement setup that required less operator intervention in autumn 2015, as described above.

### 3 Method

#### 3.1 Measurement setup

The measurements follow the TCCON standard settings wherever possible. A solar (lunar) tracker is mounted on the roof of
the AWIPEV observatory and the light is reflected into the laboratory underneath and into the FTIR spectrometer. Accurate tracking is ensured by usage of a four-quadrant diode with feedback to the solar tracker motor controller. The incident light


is focused on an entrance aperture and afterwards parallelised to enter a Michelson interferometer arrangement of the Bruker IFS 120-5 HR. The movable retro-reflective mirror is mounted on a sledge on steel rods. Accurate tracking of the movable mirror's position is provided by a stabilized internal HeNe laser reference. The light path arrives in the detector compartment of the instrument, where it is focused through a HeNe laser filter onto the InGaAs detector. The resulting signal is amplified
and recorded together with the internal laser reference.

In a post processing step the spectra are calculated via a Fast Fourier Transform (FFT) routine by the instrument operating software OPUS (by Bruker). After changing the measurement routine in 2015 to a semi-automated setup, less intervention from the operator is required. At the same time, the interferograms are read directly from the instrument, resulting in raw data slices that are processed to spectra via the i2s program shipped with the GGG2014 software suite used within TCCON.

Differences to the solar, TCCON measurements and post processing setup are: the detector itself, the spectral resolution, integration time and the size of the entrance aperture. Generally speaking, decreasing the resolution allows for integration of more interferograms in the same time frame and the larger the entrance aperture, the more light is incident on the detector, i.e. the signal. The impact of spectral resolution is further discussed in section 3.4. The entrance aperture was set as large as possible (typically a diameter of $3.15\,\mathrm{mm}$), given full illumination to ensure a uniform light path in the instrument. The actual
setting depends on lunar phase, as a smaller lunar cross-section requires a smaller aperture.

### 3.2 Calculation of dry-air mole fractions

For this analysis the current TCCON standard processing code GGG2014 was used for both solar and lunar retrievals. The retrieval code returns vertical columns ($\mathrm{VC_{gas}}$), that have to be converted to dry-air mole fractions. There are two possibilities to do this. The standard TCCON processing uses the simultaneously retrieved vertical $O_2$ column to scale the target gas'
vertical column via:

$$\mathrm{xGas} = \frac{\mathrm{VC_{gas}}}{\mathrm{VC_{O_2}}} 0.2095 \tag{1}$$

The dry-air mole fraction of $O_2$ is well known and assumed constant; therefore systematic errors common to both vertical column retrievals cancel out using this approach.

However, for the retrieval of $O_2$ the spectral band at $1.27\,\mathrm{\mu m}$ ($7880\,\mathrm{cm^{-1}}$) is used and the detector is much less sensitive in
that region compared to the $CO_2$ and $CH_4$ windows between $5800\,\mathrm{cm^{-1}}$ and $6400\,\mathrm{cm^{-1}}$ (compare Fig. 1). This results in a noisier $O_2$ retrieval especially under low signal-to-noise conditions (see Fig. 2).

The second option to calculate the dry-air mole fraction involves the scaling to atmospheric surface pressure and a correction for the water contained in the column:

$$\mathrm{xGas} = \frac{\mathrm{VC_{gas}}}{\frac{p_0 \mathrm{N_A}}{m_{dry}^{air} \bar{g}} - \mathrm{VC_{H_2O}} \frac{m_{\mathrm{H_2O}}}{m_{\mathrm{dry}}^{air}}} \tag{2}$$





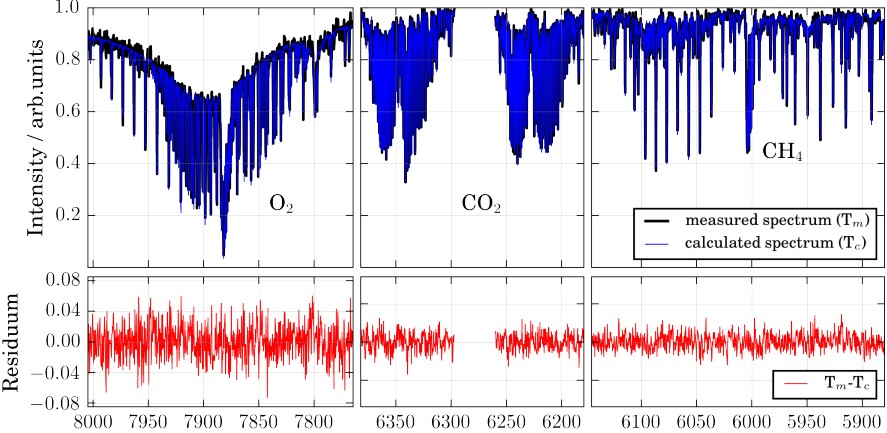

**Figure 2.** Example fit of a measured spectrum (black line) on October 25 2015, the corresponding calculated spectrum (blue line) and their residuum (red line) for the retrieved windows of O2, $CO_2$ and $CH_4$.

Here, xGas denotes the target species' dry-air mole fraction, $VC_{gas}$ the vertical column and $p_0$ the surface pressure. $N_A$ is Avogadro's number and the molecular masses of water, $m_{H_2O} = 18.01534\,\mathrm{g\,mol^{-1}}$, and dry air, $m_{dry}^{air} = 28.9644\,\mathrm{g\,mol^{-1}}$, are given. $\bar{g}$ denotes the column averaged gravitational acceleration at the measurement site and is assumed to be $\bar{g} = 9.81\,\mathrm{m\,s^{-2}}$.

This approach requires accurate knowledge of the surface pressure $p_0$. Additionally systematic errors, e.g. pointing errors

can affect the retrieval, as they are not cancelled out via ratio with $O_2$. The surface pressure measurement is performed at the Ny-Ålesund station of the Baseline Surface Radiation Network (BSRN), located adjacent to the AWIPEV observatory and thus the FTIR spectrometer. The raw pressure measurements are then scaled to compensate for the height difference to the FTS. The meteorological data is provided by AWIPEV and publicly available at doi:10.1594/PANGAEA.150000 for years until 2013, with corresponding updates for more recent years.

In the following, the approach described in equation 1 was used to retrieve $xCO_2$ and $xCH_4$. The second approach, in equation 2, was only used to derive $xO_2$ for the analysis in section 4. The main retrieval windows and the fit residuals of an example spectrum are shown in Fig. 2. The vertical column of $H_2O$ used for the water correction in equation 2 is retrieved simultaneously in several micro-windows in the same spectral region as the target species.

### 3.3 Atmospheric model

Information on the target gas is retrieved from the processed spectra by the least-square fitting algorithm GFIT (see Sec. 3.2). The software assumes an a priori profile of the target gas and calculates an artificial spectrum given additional information on the atmospheric profile. In TCCON the interpolation of the NCEP/NCAR reanalysis data (NCEPNCAR, 2016) to the sites latitude, longitude and local noon is used as an atmospheric model, resulting in one model profile per day. The NCEP/NCAR reanalysis data is publicly available and was provided via http://www.esrl.noaa.gov/psd/ (NCEPNCAR, 2016). In case of





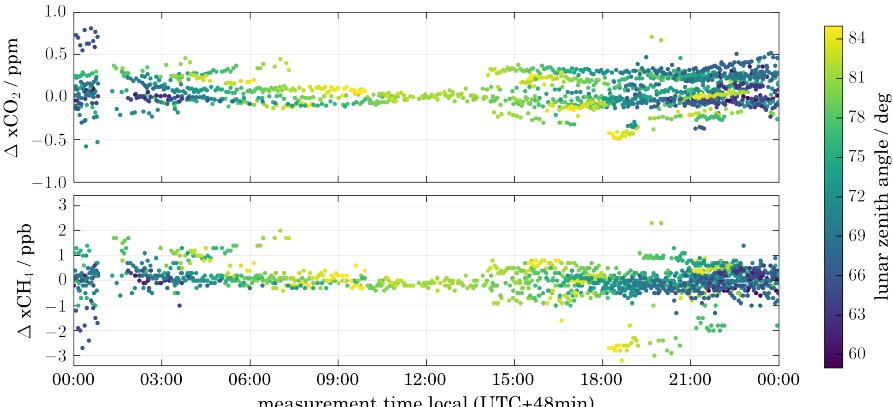

**Figure 3.** Differences in the lunar absorption retrieval results (2012 – 2015) using the site and time of measurement interpolated atmospheric model compared to using the model interpolated to site and local noon for both target species dependent on the lunar zenith angle.

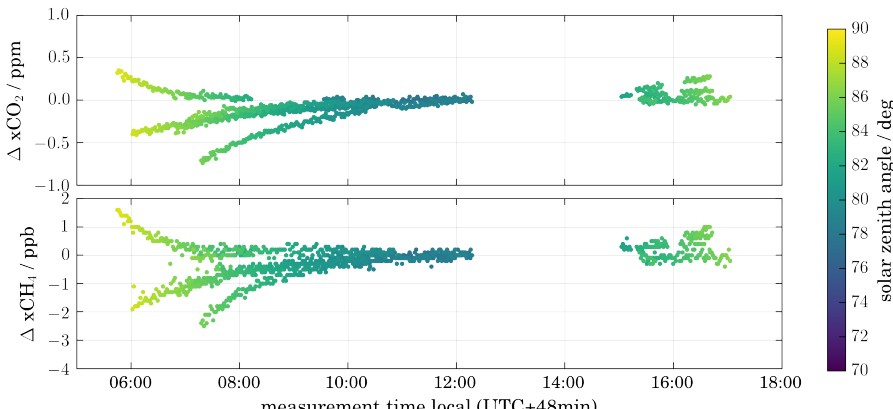

**Figure 4.** Same as Fig. 3 but for TCCON solar absorption measurements for the time between 2013-09-19 and 2013-09-24. Note the generally higher differences at high zenith angles. Between 12:30 and 15:00 local time the sun moves behind a mountain at lower zenith angles.

lunar measurements, this presents a potential problem around midnight, as consecutive measurements would use different atmospheric models, i.e. the one interpolated to local noon.

Given that the reanalysis data are available in six hour time intervals, we use the model profile interpolated to the site coordinates and the time of measurement, resulting in specific model profiles for each measurement. These profiles presumably better reflect the atmospheric conditions, especially at night. The increased computational effort for this per-spectrum-model approach is affordable for this comparatively small time series.

A comparison of the differences in retrieved $xCO_2$ and $xCH_4$ between the daily and spectrum-specific model profiles is shown in Fig. 3 for the lunar time series and for selected days in the TCCON time series in Fig. 4. The two retrievals show





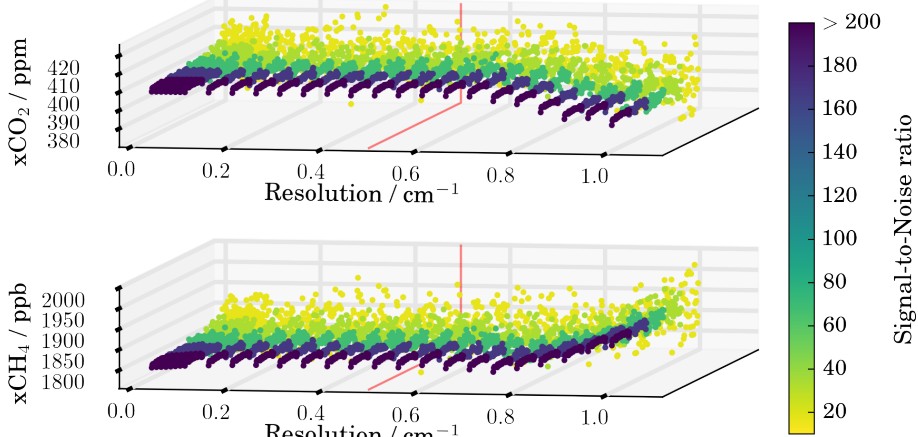

**Figure 5.** Retrieved $xCO_2$ and $xCH_4$ from cropped interferograms with different resolutions and different levels of white noise (z-axis and colorbar) added to the spectra.

minimal differences at local noon (as they should), but differences of about $\pm 0.5$ ppm (CO2) and about $\pm 2$ ppb (CH4) can occur later in the day, under quickly varying atmospheric conditions distant in time from local noon.

### 3.4 Analysis of optimal resolution

The resolution used in the TCCON is better than $0.02$ cm$^{-1}$, corresponding to a maximum optical path difference (OPD)
of $45$ cm. Initial tests showed that even with the cooled detector, the spectral signal-to-noise ratio did not allow for a robust retrieval unless a lot of spectra were averaged; however, the path of moonlight through the atmosphere changes rapidly with time. Although this is more prominent in lower latitudes, it still must be considered here, especially at large lunar zenith angles. To avoid bias from inaccurate knowledge of the viewing geometry, the integration time per measurement must be as small as possible.

One option to decrease the measurement time is to increase the velocity of the instrument's scanning mirror; however previous studies (e.g. Messerschmidt et al. (2010)) have linked an increase in scanner velocity to the increase of laser sampling errors. The scanner velocity was therefore not changed and kept at $10$ kHz to minimise potential differences from the solar absorption measurements. The second option is to decrease the spectral resolution, which increases the spectral signal-to-noise ratio. Additionally, it allows for shorter measurement times and thus for more spectra to be averaged within the same time,
resulting again in an increased signal-to-noise ratio.

The influence of resolution on the retrieval can be analysed in further detail and to circumvent differences arising from a varying atmospheric state. Previously, Petri et al. (2012) investigated this for the TCCON standard retrieval windows. Here the analysis was repeated with emphasis on lower resolutions (down to $1.0$ cm$^{-1}$) and additionally spectra with different signal-to-noise ratios were used.




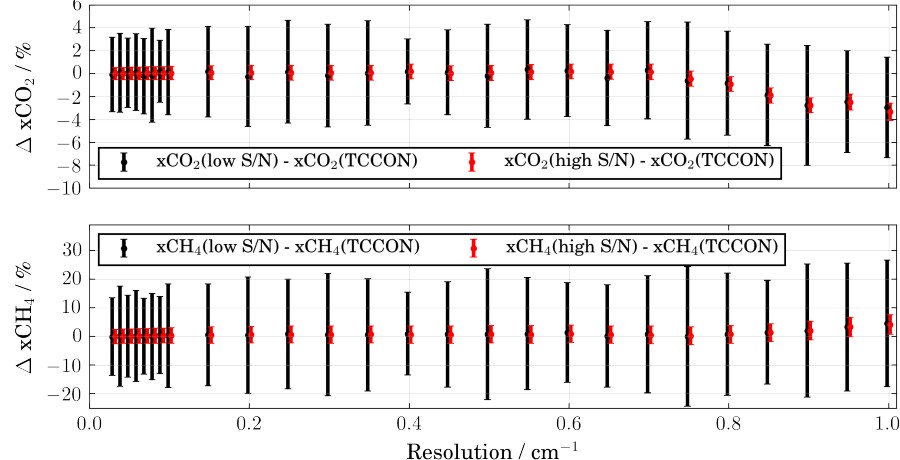

**Figure 6.** Mean of the retrieved $xCO_2$ and $xCH_4$ from cropped interferograms at different resolutions with low and high signal-to-noise ratio. Shown is the relative difference to the highest signal-to-noise ratio and highest resolution. Errorbars show the $1\sigma$ standard deviation of the mean.

A set of 60 consecutive solar spectra has been selected and the interferograms cropped at lengths corresponding to a range of maximum optical path differences between 45 cm ($0.02\ \mathrm{cm}^{-1}$) and 0.9 cm ($1.0\ \mathrm{cm}^{-1}$). The interferograms were reprocessed and the spectra calculated with the i2s program within the GGG2014 program suite.

In addition to this series of spectra, different magnitudes of white noise were added to the created spectra to simulate the effect of the lower signal-to-noise ratio expected in lunar spectra. The signal-to-noise-ratios are calculated from the reprocessed spectra by dividing the maximum mean signal between absorption lines at about $6000\ \mathrm{cm}^{-1}$ by the root mean square of a blacked out region of the spectrum. Figure 5 shows the results of the standard retrieval of $xCO_2$ and $xCH_4$ for the various combinations of resolution and signal-to-noise ratio of the series.

For better visibility, Fig. 6 shows a subset of the data from Fig. 5, showing the mean retrieved $xCO_2$ and $xCH_4$ DMFs at a given resolution. The errorbars give the standard deviation ($1\sigma$) of the arithmetic mean. Two series have been selected, with high (red) and low (black) signal-to-noise ratios. A distinct cut-off above $0.7\ \mathrm{cm}^{-1}$ can be identified in the $xCO_2$. For higher resolutions, i.e. $0.02 - 0.7\ \mathrm{cm}^{-1}$ no significant difference is visible in high signal-to-noise conditions. In general, a lower signal-to-noise ratio of the spectra leads to increased scatter of the retrieved DMFs, but to no significant bias. Table 1 shows the bias in the retrieved DMFs of high and low signal-to-noise ratio spectra for the two resolutions used in the measurement setup later.

Gisi et al. (2012) showed that lower resolution solar spectra can be used to retrieve DMFs with a low resolution FTS (Bruker EM27/SUN). Recently Hedelius et al. (2016) investigated errors and biases from a $0.5\ \mathrm{cm}^{-1}$ FTS (Bruker EM27) for TCCON relevant species. The three studies (Petri et al., 2012; Gisi et al., 2012; Hedelius et al., 2016) report different biases in $xCO_2$ when changing the resolution to $0.5\ \mathrm{cm}^{-1}$ in the range from $-0.12\ \%\ \%$ to $0.13\ \%$. For $xCH_4$, Hedelius et al. (2016) reported





**Table 1.** Comparison of the biases, introduced by lower resolution measurements and low signal-to-noise ratio. Subset of data points from Fig. 6

| S/N | Resolution [cm$^{-1}$] | $\Delta$xCO$_2$ [%] | $\Delta$xCH$_4$ [%] |
|---|---|---|---|
| > 300 | 0.08 | 0.03 $\pm$ 0.57 | 0.28 $\pm$ 2.61 |
| | 0.5 | 0.07 $\pm$ 0.65 | 0.76 $\pm$ 3.03 |
| $\approx$ 30 | 0.08 | -0.13 $\pm$ 4.12 | 0.00 $\pm$ 15.03 |
| | 0.5 | -0.20 $\pm$ 4.50 | 0.79 $\pm$ 22.89 |

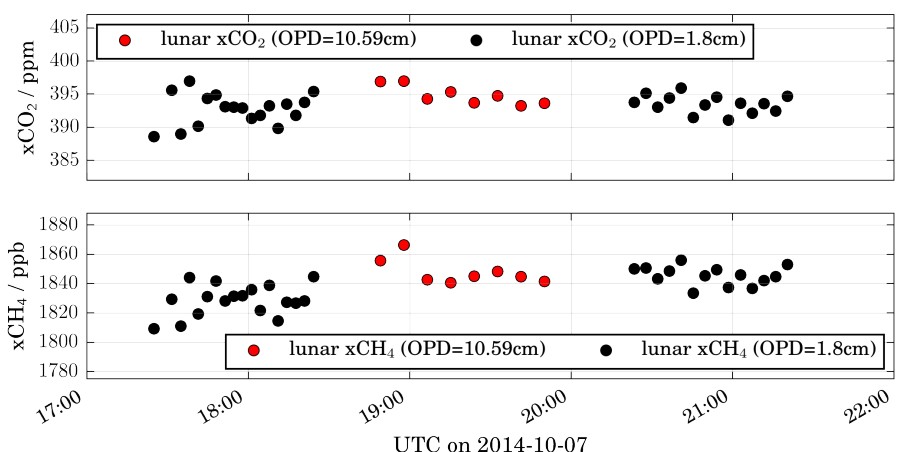

**Figure 7.** Comparison of retrieved xCO$_2$ and xCH$_4$ for different resolutions from low (**OPD** = 1.8 cm $\widehat{=}$ 0.5 cm$^{-1}$, black) and higher (**OPD** = 10.59 cm $\widehat{=}$ 0.085 cm$^{-1}$, red) resolution measurements on 2014-10-07.

an increase of 0.28 %, when decreasing the the resolution to 0.49 cm$^{-1}$. In our analysis (see Tab. 1) a consistent decrease in mean $\Delta$xCO$_2$ and $\Delta$xCH$_4$, i.e. the difference between DMFs from low and high resolution spectra, is observed when moving to lower resolutions. However, when considering the assigned errors (1$\sigma$ standard deviation) this is not significant, especially under lower signal-to-noise conditions.

5    For the final decision on the best resolution for low S/N conditions the possible number of recorded spectra per time interval has to be considered. This number does not increase linearly, due to instrumental effects, i.e. the deceleration of the moving mirror and the time needed for data acquisition and storage. The first measurements were taken at a reasonably high spectral resolution of 0.08 cm$^{-1}$ (OPD = 11.25 cm). The measurement setup was adjusted after further tests. The benefit of a better signal-to-noise ratio on the measurement precision lead to finally decreasing the resolution to 0.5 cm$^{-1}$ (OPD = 1.8 cm) and

10   all measurements from 2015 onwards were taken with a resolution of 0.5 cm$^{-1}$.



The effect of different resolutions on the retrieved columns can also be investigated by comparing different measurements taken consecutively with different resolutions. Figure 7 shows lunar absorption measurements of the target species on October 7 2014. The first and third batch of measurements were taken with a resolution of $0.085\ \mathrm{cm}^{-1}$ (OPD = 10.59 cm), the second batch was measured with $0.5\ \mathrm{cm}^{-1}$ (OPD = 1.8 cm) resolution. No significant bias is observed.

Decreasing the spectral resolution also changes the information content of the recorded spectral lines. This results in a change in shape of the measurements averaging kernels and is discussed below.

## 3.5   Averaging Kernels

The sensitivity of the retrieved dry-air mole fraction of the target gas depends on the a priori information and the measurement's altitude dependent sensitivity, i.e. the averaging kernels. The a priori profiles used are the default TCCON ones. The averaging

kernel of a measurement depends on the viewing geometry as well as the resolution, the absorption strength and the signal-to-noise ratio. The averaging kernels strongly depend on the information content of the spectrum. The weight different altitude levels have in the retrieval can be parameterized as a function of the zenith angle. As the instrument faces the light source at a certain zenith angle, the measurement samples different contributions from the various atmospheric layers. The pressure broadening of the absorption features shows a specific altitude dependent sensitivity and this information depends on the chosen

resolution and the signal-to-noise ratio of the measurement.

    The setup of the lunar measurements is similar to that of TCCON measurements, therefore the averaging kernels are quite similar, aside from effects of resolution and noise for a given zenith angle.

    The top panel in Fig. 8 shows the averaging kernels for the lunar measurements. The middle panel shows the difference from the standard TCCON ones from Ny-Ålesund, interpolated to the corresponding zenith angles. The lines are color gradient

coded with their respective zenith angles and different color schemes reflect different resolutions.

    Pressure broadening leads to spectral lines originating from gases at low pressure being narrower than those at higher pressure. The narrow part of a spectral line sampled with fewer points therefore cannot give as much information as one with higher resolution. This leads to averaging kernels from low resolution spectra being less sensitive to the stratosphere and more sensitive in the lower troposphere than their high resolution counterparts. This can be seen in the lower panel of Fig. 8, where

the difference between standard TCCON averaging kernels and their lower-resolution counterparts at the same zenith angle is shown. As expected, decreasing the spectral resolution leads to greater differences between the averaging kernels.

## 4   Validation with solar absorption spectroscopy

The validation of the measurements performed during the polar night is difficult. In the absence of other options, here we compare to solar absorption measurements taken within TCCON. In spring and autumn there are a few consecutive days

around the full moon where solar absorption measurements during the day and lunar absorption measurements during the night are possible. Such comparison measurements were performed in March and September 2013. Here the DMFs of $xCO_2$ and



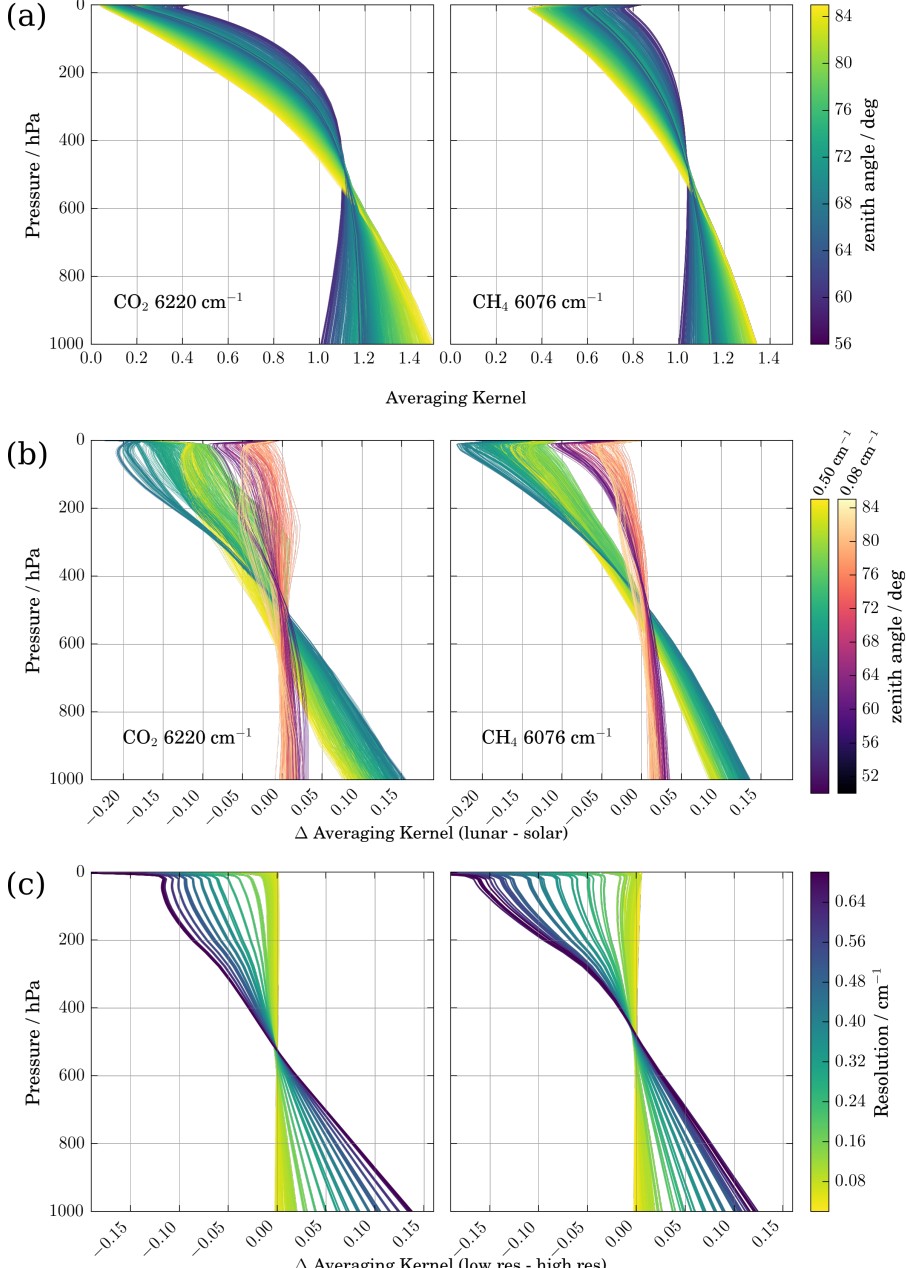

**Figure 8.** (a) Averaging kernels of the lunar measurements. (b) Difference between lunar and solar averaging kernels color coded for different spectral resolutions. (c) Differences between low resolution and TCCON spectra averaging kernels as a function of resolution.

$xCH_4$ for both solar and lunar measurements were retrieved using equation 1. For the comparison of $xO_2$ equation 2 was used, respectively.





**Table 2.** Comparison of the retrieved solar, lunar and model DMFs for the two comparison time periods. Note that $xO_2$ was calculated using the surface pressure and the offset to the true atmospheric value of 20.95% is caused by spectroscopic errors.

|  |  | $xCO_2$ [ppm] | $xCH_4$ [ppb] | $xO_2$ [%] |
|---|---|---|---|---|
| March 2013 | solar | $397.47 \pm 0.67$ | $1773.78 \pm 2.99$ | $21.33 \pm 0.08$ |
|  | lunar | $396.81 \pm 3.89$ | $1775.72 \pm 17.64$ | $21.34 \pm 0.36$ |
|  | Jena $CO_2$ | $398.01 \pm 0.13$ | – | – |
|  | CT15 $CO_2$ | $396.89 \pm 0.22$ | – | – |
|  | MACC $CO_2$ | $397.16 \pm 0.18$ | – | – |
|  | MACC $CH_4$ | – | $1784.09 \pm 1.06$ | – |
| September 2013 | solar | $393.16 \pm 0.49$ | $1810.26 \pm 3.11$ | $21.38 \pm 0.06$ |
|  | lunar | $392.15 \pm 8.03$ | $1813.62 \pm 38.02$ | $21.40 \pm 0.60$ |
|  | Jena $CO_2$ | $391.56 \pm 0.26$ | – | – |
|  | CT15 $CO_2$ | $391.29 \pm 0.24$ | – | – |
|  | MACC $CO_2$ | $392.07 \pm 0.39$ | – | – |
|  | MACC $CH_4$ | – | $1800.79 \pm 1.58$ | – |

Assuming the total column values do not change significantly during that time period, the means of the two retrievals can be compared directly. Figure 9 shows the comparison results and the calculated means for a comparison in September 2013. Table 2 shows the corresponding values of the arithmetic mean and its standard deviation as an indication of the error for both comparison campaigns in March and September 2013. The same analysis was performed on the available smoothed

model output. The calculated standard deviation of the models of about $0.2\,\mathrm{ppm}$ (March) and $0.3\,\mathrm{ppm}$ (September) for $CO_2$ and $1.0\,\mathrm{ppb}$ and $1.6\,\mathrm{ppb}$ respectively for $CH_4$ indicates that the assumption of stable DMFs for the observed time frame is reasonable. During this brief comparison time period, no significant bias can be observed when comparing the lunar with the solar DMF retrievals.

As described in section 3.2 (see equation 1), the dry-air column is calculated using the vertical column of $O_2$, retrieved from

10 the $7885\,\mathrm{cm^{-1}}$ spectral region. Here air-glow emissions in the high atmosphere could potentially disturb the $O_2$ spectra. This can typically be ignored in solar absorption spectra, as the magnitude of the emissions is negligible, when viewing directly into the sun. In case of lunar spectra, however, air-glow emissions could potentially fill in the spectral lines and influence the measurements. To test this, $xO_2$ was retrieved using the surface pressure to calculate the dry-air column as described in equation 2.

In both comparison periods, no significant difference between the solar and lunar retrievals of $xO_2$ can be observed. Note that $xO_2$ retrieved via surface pressure shows an offset of $0.4\,\%$ in both cases (lunar and solar). This offset originates in the line parameters used for the $O_2$ retrieval and is compensated in the $xCO_2$ and $xCH_4$ retrieval with the TCCON in-situ correction. Washenfelder et al. (2006) reported values that are $2.27 \pm 0.25\,\%$ larger if the surface pressure retrieved dry column was used.





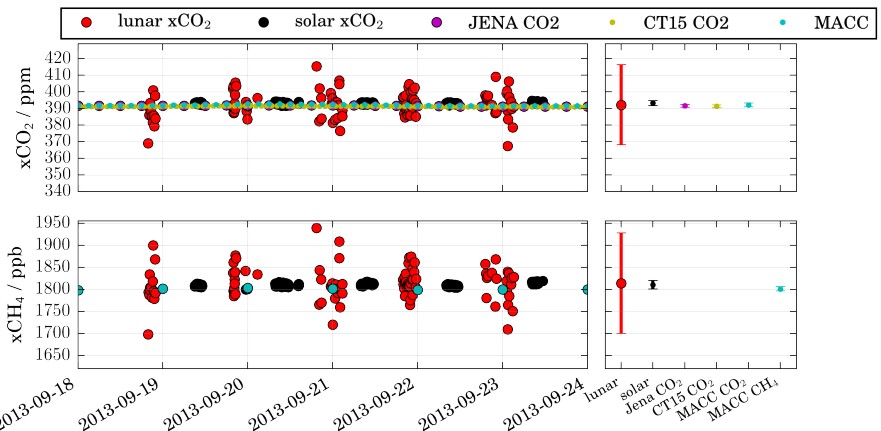

**Figure 9.** Comparison of the solar and lunar measurements of $xCO_2$ and $xCH_4$ in September 2013 (dots) and the corresponding arithmetic means (lines). Values are given in Table 2.

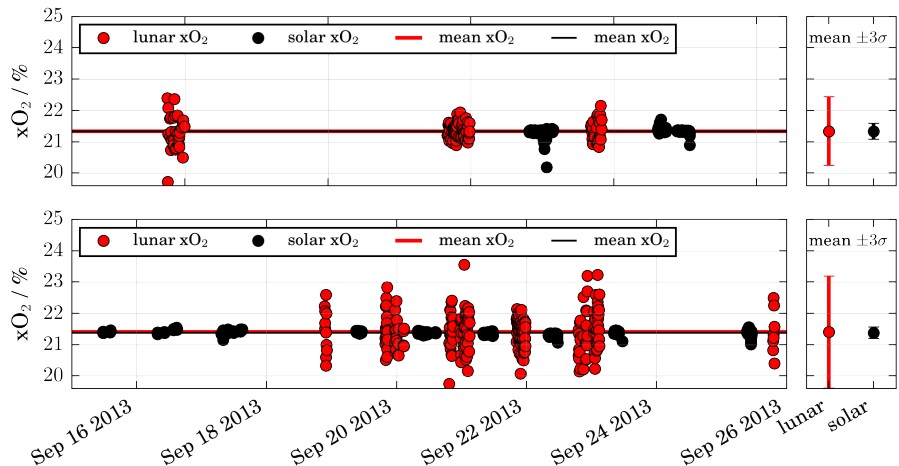

**Figure 10.** Comparison of the solar and lunar measurements of $xO_2$ in March and September 2013.

Here we find a mean difference of $1.96 \pm 0.14\,\%$, when calculating the mean and standard deviation of the solar and lunar mean $xO_2$ values shown in the sidebars in Fig. 10. Note that these retrievals were performed with updated spectroscopy available within GGG2014 compared to that used by Washenfelder et al. (2006).





## 5 Seasonal cycle and model comparison

### 5.1 Method – model comparison

The rigorous comparison of ground-based column measurements of a trace gas to model simulations requires resampling the model profile as if it was measured by the instrument.

The smoothed column dry-air mole fraction $\hat{c}$ can be calculated following Rodgers and Connor (2003); Connor et al. (2008); Wunch et al. (2010) by adding the column integrated a priori profile ($c_a$) to the difference between the model ($\mathbf{x}$) and the dry TCCON a priori profile ($\mathbf{x_a}$) weighted with the averaging kernel ($\mathbf{a}$):

$$\hat{c} = c_a + \mathbf{h}^T \mathbf{a}^T (\mathbf{x} - \mathbf{x}_a) \tag{3}$$

Here, $\mathbf{h}$ represents the pressure weighting function (see Connor et al. (2008)).

Given a vertical model profile, the measurement's averaging kernel and the vertical columns of water vapour and the a priori profile of the target gas, the smoothed dry-air mole fraction of the model output can be calculated. Due to the high random error of the lunar FTS measurements, daily means have been calculated for both the measurements and the model data, after the smoothing was applied.

### 5.2 Results – time series

In this section the FTIR time series is compared to $CO_2$ model results from three different models: the MACC $CO_2$ model version 14r2 (MACCCO2, 2016), the CarbonTracker 2015 (CT2015, 2016) model and the Jena $CO_2$ inversion version s04_v3.7 (JenaCO2, 2005). In case of the $CH_4$ time series, the MACC $CH_4$ v10 (MACCCH4, 2016) is used. As described in Section 5.1 the models DMF profile has been smoothed with the corresponding a priori and averaging kernel of the lunar and solar measurement, respectively. For times where there are no FTS measurements available, an averaging kernel was calculated
using the solar zenith angle of the corresponding time. In winter the lunar zenith angle was used instead. For times where no FTS measurements were possible at all, e.g. sun and moon are below the horizon, a mean zenith angle of $65°$ was assumed.

The resulting model time series can now be compared directly to the FTS measurements. Figure 11 shows the comparison of the FTS and the smoothed model time series for $CO_2$. The $CH_4$ comparison is shown in Fig. 12.

### 5.3 Results – seasonal cycle

The seasonal cycles of the target species are similar from year to year, except for a secular increase in both, $xCO_2$ and xCH4. In the following the de-trended seasonal cycles are compared to the models already discussed in section 5.2.

Figure 13 shows the seasonal cycle of $xCO_2$ as observed with the Ny-Ålesund FTS between 2012 and 2016, detrended with a linear increase of $2.6\,\mathrm{ppm\,yr^{-1}}$, an offset of $380.0\,\mathrm{ppm}$ on 2012-01-01 and condensed to one year. The seasonal cycle of $xCO_2$ shows little difference between the three models, therefore the comparison can be performed with an model average.
The shaded area in Fig. 13 shows the $3\sigma$ standard deviation around the daily mean of the combined model data points of





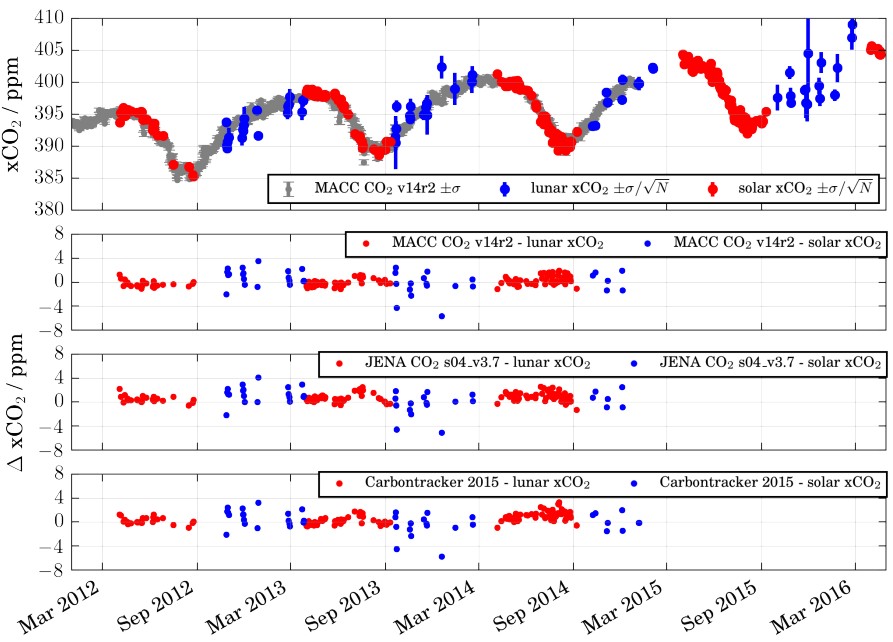

**Figure 11.** Comparison of the daily means of lunar (blue) and solar (red) $xCO_2$ FTIR measurements to the AK-smoothed MACC $CO_2$ model v14r2 (top panel, gray). Errorbars show the standard error ($\sigma/\sqrt{N}$, with $N$ number of measurements). The lower panels show the difference model - measurement for all models.

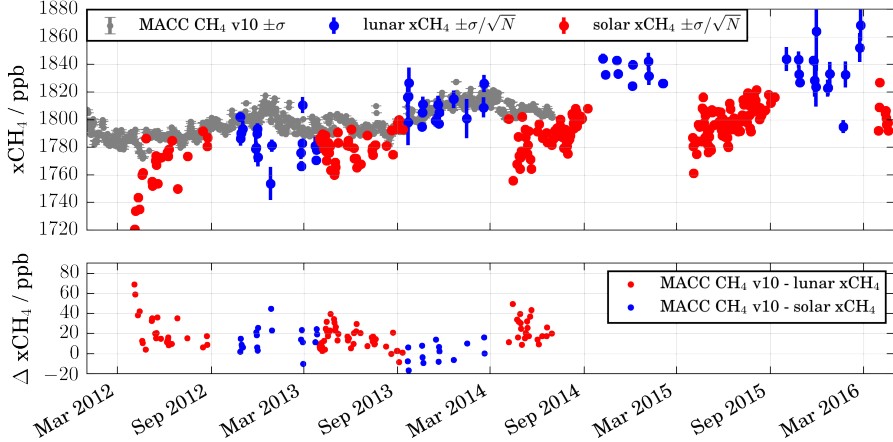

**Figure 12.** Comparison of the daily means of lunar (blue) and solar (red) $xCH_4$ FTIR measurements to the AK-smoothed MACC $CH_4$ model v10 (gray). Errorbars show the standard error ($\sigma/\sqrt{N}$, with $N$ number of measurements).The lower panel shows the difference model - measurement.





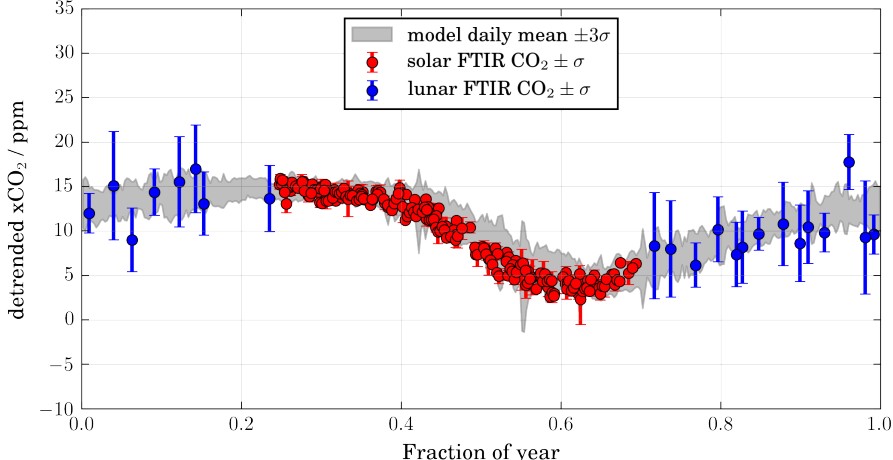

**Figure 13.** Comparison of solar and lunar $xCO_2$ FTIR measurements (green dots). Errorbars show $1\sigma$ standard deviation of the daily mean. The lunar data points have been averaged over one full moon period each. The shaded gray area shows the $1\sigma$ standard deviation of the three model daily means (MACC, CarbonTracker and Jena) as shown in Fig. 11.

all three models (MACC, CarbonTracker and Jena). The weighted average of all FTS measurements during one full moon period is shown (green dots) with errorbars corresponding to the standard error ($\sigma/\sqrt{N}$) of the daily mean calculated from $N$ measurements. The weights are chosen to be the inverse squared residual of the spectral fits.

The difference between the models and the TCCON measurements in summer is quite small, except for a phase shift in the onset of the downward slope at the beginning of the growing season decline. In winter the models agree well with the FTIR lunar absorption measurements, within the given error margin.

In the case of $CH_4$ a similar comparison has been performed and the results can be seen in Fig. 14. Here the $xCH_4$ time series have been linearly detrended with an annual increase of $10.6 \ \mathrm{ppb \, yr^{-1}}$ and an offset of $1760.0 \ \mathrm{ppb}$ on 2012-01-01. Figure 14 shows the $3\sigma$ standard deviation around the daily means of the MACC CH4 model (shaded area) compared to the FTS measurements (green dots) averaged over one full moon measurement cycle. The errorbars correspond to the $1\sigma$ standard deviation of the mean.

In spring/summer the FTS measurements show generally smaller values than the model and a larger spread. From late summer throughout the winter the measurements are in better agreement with the model.

## 6 Conclusions

Measurements of the column averaged dry-air mole fractions of $CO_2$ and $CH_4$ have been performed in the polar night from 2012 to 2016 to complement the established solar absorption measurements within the TCCON. The newly employed thermo-





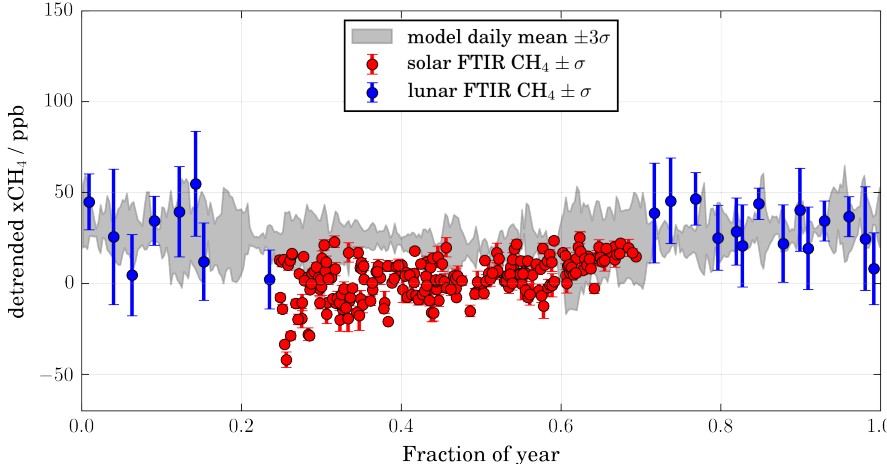

**Figure 14.** Comparison of solar and lunar $xCH_4$ FTIR measurements (green dots). Errorbars show $1\sigma$ standard deviation of the daily mean. The lunar data points have been averaged over one full moon period each. The shaded gray area shows the $1\sigma$ standard deviation of the MACC CH4 model daily means as shown in Fig. 12.

electrically cooled InGaAs detector allows the usage of reflected sunlight on the full lunar disc to serve as a light source above the atmosphere to perform lunar absorption spectroscopy in the near-infrared spectral region.

The lunar absorption measurements have been validated with standard TCCON measurements in spring and autumn 2013 and the comparison shows no significant biases. The decrease of spectral resolution allows for an increase of the spectral signal-to-noise ratio, which in turn decreases the random error significantly. Under optimal conditions, lunar measurements with standard deviation of the daily mean ($1\sigma$) of about 2 ppm for $xCO_2$ and about 10 ppb for $xCH_4$ can be achieved using this approach. This corresponds to a precision of about 0.5 % for each gas.

The newly created time series has been compared to different model simulations. All three $CO_2$ models (MACC CO2 model v. 14r2), CarbonTracker 2015, Jena $CO_2$ inversion s04_v3.7) are generally in good agreement with the FTIR measurements. The $xCH_4$ time series shows large deviations in spring/summer and an overall good agreement in autumn/winter.

## 7  Data availability

The TCCON data are publicly available from the TCCON archive at http://tccon.ornl.gov/. The solar and lunar measurement data used in this study has been uploaded to the Pangaea database and is available at https://doi.pangaea.de/10.1594/PANGAEA.872007. The MACC model data can be accessed via http://www.gmes-atmosphere.eu/. The Jena $CO_2$ inversion is provided via http://www.bgc-jena.mpg.de/CarboScope and the Carbontracker data can be found at http://carbontracker.noaa.gov. The NCEP/NCAR reanalysis is provided by NOAA via http://www.esrl.noaa.gov/psd/. The surface meteorology data from the Ny-Ålesund BSRN station used here is available from doi:10.1594/PANGAEA.150000.



*Author contributions.* The measurements where taken by M.Buschmann, the co-authors and the AWIPEV station staff. M.Buschmann performed the analysis and prepared the manuscript with contributions from all co-authors.

*Competing interests.* The authors declare that they have no conflict of interest.

*Acknowledgements.* This project was funded from the German Research Foundation (DFG) as part of the grant NO 404/17. M.Buschmann

5    received additional funding from the Helmholtz Earth System Science Research School (ESSReS), through the SFB/TR172 "ArctiC Amplification: Climate Relevant Atmospheric and SurfaCe Processes, and Feedback Mechanisms (AC)$^3$" by the DFG and the GAIA-CLIM Horizon-2020 project of the European Union. The authors would like to thank the station personnel of the AWIPEV station, who operate the instrument on a routine basis. We further thank the AWI staff at Potsdam for providing the long term observations of surface meteorology and we thankfully acknowledge the provision of NCEP Reanalysis data by the NOAA/OAR/ESRL PSD, Boulder. The FTIR Observations

10   in Ny-Ålesund are funded by the EU projects InGOS and ICOS-INWIRE, and by the Senate of Bremen. Nicholas Deutscher is supported by an ARC-DECRA, DE140100178. We thank Stanley Sander at JPL for initial discussions on the detector design. Additionally we would like to thank Frederic Chevallier at LSCE for providing the MACC model data and for helpful comments on an earlier version of the paper.



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
