# Peer review of "The arctic seasonal cycle of total column $CO_2$ and $CH_4$ from ground-based solar and lunar FTIR absorption spectrometry"

_Atmospheric Measurement Techniques, 2017_

## Referee Comment (RC1) · Anonymous Referee #1 · 27 Feb 2017

The manuscript under consideration discusses measurements of column-averaged amounts of carbon dioxide and methane by applying near-infrared ground based lunar absorption spectroscopy. The suggested approach might be regarded as a useful addition to solar measurements especially at high latitude sites, because then the sun is inaccessible during longer periods in wintertime. However, in my feeling, more emphasis should be laid on potential systematic biases of lunar measurements (the abstract only provides an estimate for the precision!). Moreover, a crucial question is left untreated: which quality (precision and accuracy) is required for such observations in order to justify the associated effort and to improve our knowledge of the atmospheric state? The de-trended wintertime year-to-year variability of the MACC data would al-

low estimating figures. One potential source of systematic bias is due to the fact that the solar spectrum observed on the moon is a solar disc-averaged spectrum, while TCCON observes disc-centered spectra (and this is assumed in the GFIT analysis also, therefore the lunar spectra need to be processed with different settings). A discussion of the two crucial items (accuracy budget and of the target accuracy and precision) should be discussed in the final version of the paper. My detailed technical comments are given below.

Abstract: also provide an estimate for the accuracy (bias with respect to solar TCCON measurements) of the lunar measurements.

Page 2, line 28: "The extension of the bandgap . . . reduces the quantum efficiency" – is this true?

Figure 1: It would be instructive to show a lamp spectrum recorded with the standard TCCON detector element also (and to provide some information concerning the noise level achieved with the selected lunar InGaAs diode (cooled and uncooled) and with the standard extended TCCON detector element for the same input signal level, e.g. for the 6000 . . . 6400 cm-1 region, where the CH4 and CO2 bands reside).

Page 4, line 10 ff: not a sentence.

Section 3.2: The fact that the noise level is too high in the lunar observations for using the spectroscopically observed oxygen column should be regarded as a severe drawback of the suggested approach.

Page 5, line 11: ". . .for the analysis in section 4." It would be instructive to explain here to the reader which topic is covered in Section 4.

Page 7, line 10 ff: ". . .one option to decrease measurement time . . . is to increase the velocity . . .". No, this is not the case in the context of optimizing the spectral signal-to-noise-ratio (SNR). Here, only the spectral resolution and the throughput matter.

I would have expected (for a given allowed integration time) to see a more pronounced

reduction of error bars on the retrieved columns until a further reduction of resolution starts to decrease the contrast between the lines and the adjacent continuum. Has the spectral SNR been adjusted as function of resolution in this manner (assumption of a certain amount of available integration time)? When comparing different resolutions, one might also take into account that a larger fieldstop can be applied when resolution is reduced (increasing the signal level, favoring shorter scans even more).

Page 8, line 4: "...white noise were added.". How has this operation been performed technically? In the interferogram domain before the FFT? Note that this section does not specify (nor treats) the choice of the numerical apodization function, which seems a further important choice in addition to the scan length if reaching the best possible precision of the retrieved column is so crucial.

Page 12, line 10: "air-glow emissions". This study would be especially interesting if lunar spectra taken during twilight would be treated separately (spelling: airglow).

Figure 12: Despite the fact that no biases were discovered in the September 2013 measurements, one is left with the impression that the lunar CH4 measurements in 2015 are biased high in comparison to the solar observations.

---

## Referee Comment (RC2) · D. Wunch (Referee) · 20 Mar 2017

Buschmann_2017

This paper describes a new set up at the Ny Alesund NDACC/TCCON station to permit near infrared measurements of reflected sunlight off the moon. The authors installed a cooled InGaAs detector for this purpose (instead of the typical room-temperature InGaAs at most TCCON stations) and performed detailed analyses of the tradeoffs between spectral resolution, signal-to-noise ratio (SNR) and retrieved XCO2 and XCH4 abundances. They settled on a resolution and field stop size and recorded lunar measurements for several years. They then compared the lunar measurements to solar measurements during spring and autumn, and to models. The lunar measurements

are less precise and accurate than the solar measurements (as expected), but it appears as though, at least in the case of XCH4, that the lunar measurements may provide some interesting constraints on Arctic methane seasonal cycle amplitude.

General Comments:

The language needs tightening - some technical concepts that are specific to TCCON or Bruker 125HR instruments that may not be familiar to the wide AMT audience are glossed over and should be written in a clearer, more general way.

Night time validation with aircraft or AirCore profiles would be best, but appear to be unavailable (at least, they are not mentioned in the manuscript). Perhaps this should be mentioned in the discussion or conclusions section.

In Figure 14, you compare the XCH4 seasonal cycle from your lunar and solar measurements to the MACC model. It shows significant disagreement in summer, but not in winter, showing that the model isn't able to properly reproduce the Arctic methane seasonal cycle amplitude. Do you have any idea why? This, to me, is one of the most interesting figures/results of the paper.

Specific/Technical Comments:

P1L4: The moon isn't a light source - it's reflected sunlight off the moon.

P1L5: I don't think you mean "parallel".

P1L23: You don't need extended InGaAs detectors to measure above 5000 cm-1.

P2L18: Do you use the solar brightness fluctuation corrections for high cirrus typically employed by TCCON (embedded in I2S for DC-recorded interferograms)?

P2L22: 0.04 what units? mrad?

P2L22: This sentence may be too technical for this audience. Explain that this ME and phase error are consistent with a well aligned instrument.

L25-35: This is too technical - please explain further.

P4L11: Rework sentence beginning with "Generally speaking, . . ."

P4L14: The entrance aperture wasn't always 3.15 mm? Please explain.

P7L1: Please note that the large deviations are at very high SZA that would be filtered out in a typical TCCON filter. Could you make this plot for days with lower SZAs? Does it look the same?

P7L11: This worry no longer holds, given that Bruker has provided two solutions to the ghost problem (the laser sampling board potentiometer and the new M16 controllers with the XSM option), and TCCON provides a ghost removal procedure with I2S, as long as you measure simultaneously on another detector with a spectral range that is entirely within a single alias. In fact, I believe Bruker recommends 20kHz as their preferred scanner speed.

Fig 5: I see what you're trying to do with this figure, but I find it very difficult to read and interpret quantitatively. Perhaps you also need to show example slices through the 3D figures showing XCH4 vs resolution and XCH4 vs SNR.

Fig 6: Would the x-axis scale work better as a log10 scale? Also, with the low SNR error bars as large as they are, it's difficult to see what the mean value is as a function of resolution. Perhaps you need to reduce the y-axis limits and show a representative error bar.

P10L10: The averaging kernel also depends on the retrieval methodology.

P12L1: Can you assume that the total columns do not change significantly during the 24-hour period? What about drawdown from the terrestrial biosphere throughout the day and respiration at night? Is night time respiration a feature of the carbon cycle you can hope to measure with your lunar measurements given the precision of your measurements? The y-axis scale is too large in Figure 9 to see whether there is any diurnal cycle in your data and models. Ditto for Figure 10.
P14L25: Remove the comma after "both".

P14L25: The models don't capture the secular trends in XCO2 and XCH4? Why not?

Fig 13, 14: I don't see any green dots.

―――――――――――――――――――――

---

## Author Response (AR1)

The point-by-point overview to the referees comments taken from the author response in the manuscript discussion. Notes on what exactly was changed have been added. The page and line numbers refer to the attached LaTeX-Diff file.

**Point-by-point response to Anonymous Review #1**

General Comment:
*One potential source of systematic bias is due to the fact that the solar spectrum observed on the moon is a solar disc-averaged spectrum, while TCCON observes disc-centered spectra (and this is assumed in the GFIT analysis also, therefore the lunar spectra need to be processed with different settings).*
Response:
Sunlight reflected at the lunar surface will have a (solar-)disc-averaged spectrum, i.e. the solar lines will be broadened as a result of the different Doppler-shifted contributions from different parts of the solar disc. GFIT includes a setting, that switches to a calculation of a disc-averaged spectrum, when the moon is selected as the source, therefore no bias is expected. This will be clarified in the revised manuscript.
**Changes:**
Paragraph added on page 4 line 33-36.

General Comment:
*A discussion of the two crucial items (accuracy budget and of the target accuracy and precision) should be discussed in the final version of the paper.*
Response:
The referee is correct that the accuracy and precision of the lunar retrievals are crucial. Within the frame of this study the solar TCCON measurements are considered to be correct. Section 4 of the manuscript shows the validation with the TCCON data. Daytime TCCON data has been compared with the nighttime lunar measurements and it is assumed that diurnal variation can be neglected. We assume this is valid given that the model outputs for that time period show small variabilities in the order of 0.2 ppm (1.0 ppb) for xCO2 (xCH4) (see Tab. 2). The accuracy of the lunar measurements can be determined via the bias of the lunar compared to the solar measurements and can be deduced from Tab. 4 as well. In March 2013 the difference between solar and lunar measurements is $0.66 \pm 4.56$ ppm for xCO2 and $-1.94 \pm 20.63$ ppb for xCH4. In the September 2013 campaign a bias of $1.01 \pm 8.52$ ppm for xCO2 and $-3.36 \pm 41.13$ ppb for xCH4 can be observed. The diurnal variability of the lunar measurements is used to define the precision. As the later measurements have a higher precision, a typical value achieved in the 2014/2015 winter is used. This is given in the Conclusions as a standard deviation of the daily mean of 2 ppm (10 ppb) for xCO2 (xCH4), in both cases corresponding to about 0.5 %. This discussion will be added to Section 4 in the final manuscript and the values for the bias added to the abstract to further emphasize their importance.
The target accuracy on the other hand is more difficult to determine. As suggested by the reviewer, the detrended year-to-year wintertime variability in the models can be used as a proxy. In the smoothed, detrended MACC CO2 and CH4 model the arithmetic mean of the first week of January differs by 0.55 ppm in xCO2 and 9.84 ppb in xCH4 between 2012 and 2014. At the same time, the standard deviation of all values for the first week of January between 2012 and 2014 is about 1.8 ppm for xCO2 and 18.8 ppb for xCH4. However, these estimates are potentially subject to unknown biases in the model, i.e. the model could be biased similarly every year. Additionally, the seasonal variability surely is an upper limit for the target precision. Here the seasonal cycle amplitude measured by solar FTS is about 15 ppm for xCO2 and about 40 ppb for xCH4.
This will also be added in the revised manuscript.
**Changes:**
An extended discussion of the accuracy budget has been added on page 14 lines 4-10. A discussion of target accuracy added to pg. 14 lines 11-17.

Comment:

*Abstract: also provide an estimate for the accuracy (bias with respect to solar TCCON measurements) of the lunar measurements.*

Response:

An estimate of the accuracy as addressed in the answer to the previous comment will be added to the abstract.

**Changes:**

The abstract has been extended to include the estimates on page 1 lines 7-9.

Comment:

*Page 2, line 28: "The extension of the bandgap . . . reduces the quantum efficiency" - is this true?*

Response:

No, it should have stated "the extension of the detector sensitivity" and will be corrected in the revised manuscript. The manufacturer reports values for the noise eqivalent power (NEP), that is the power required to achieve a signal-to-noise ratio of 1, for the 1.7 $\mu$m cut-off model of 1.8E-15 $\frac{\text{W}}{\sqrt{\text{Hz}}}$ and 9.0E-15 $\frac{\text{W}}{\sqrt{\text{Hz}}}$ for the 1.9 $\mu$m cut-off model respectively. For an uncooled diode with 2.6 $\mu$m cut-off, the reported NEP is 2.1E-12 $\frac{\text{W}}{\sqrt{\text{Hz}}}$ (see `http://www.teledynejudson.com/prods/ Documents/InGaAs\_shortform\_Sept2003.pdf`).

**Changes:**

The statement has been corrected on page 2 line 31 and an extended explanation added to page 3 in lines 4-5.

Comment:

*Figure 1: It would be instructive to show a lamp spectrum recorded with the standard TCCON detector element also (and to provide some information concerning the noise level achieved with the selected lunar InGaAs diode (cooled and uncooled) and with the standard extended TCCON detector element for the same input signal level, e.g. for the 6000 . . . 6400 cm-1 region, where the CH4 and CO2 bands reside).*

Response:

As mentioned above, the sensitivities of the standard TCCON diode and the TE cooled differ by about 3 orders of magnitude. Therefore, given the vast difference, we do not feel that adding a graphical representation of this is necessary.

**Changes:**

No changes to the manuscript.

Comment:

*Page 4, line 10 ff: not a sentence.*

Response:

Sentence will be reworded to: The differences between the solar and lunar measurements include the detector, the spectral resolution, the integration time and the size of the entrance aperture.

**Changes:**

Changed on page 4 line 22.

Comment:

*Section 3.2: The fact that the noise level is too high in the lunar observations for using the spectroscopically observed oxygen column should be regarded as a severe drawback of the suggested approach.*

Response:

This is a misunderstanding. The DMFs used in this study use the O2-ratio approach, the noise here is not 'too high' but merely higher than using the surface pressure approach to calculate the DMFs. However, the error cancellation properties of the O2-ratio approach outweigh the potential

lower noise achieved by using the surface pressure. We have attempted to clarify this in the paper to avoid confusion.
**Changes:**
No changes to the manuscript.

Comment:
*Page 5, line 11: ". . .for the analysis in section 4." It would be instructive to explain here to the reader which topic is covered in Section 4.*
Response:
The sentence will be changed to: In the following, the approach described in equation 1 was used to retrieve xCO2 and xCH4. The second approach, in equation 2, was only used to retrieve xO2 in Section 4, which covers the validation with solar measurements.
**Changes:**
Extension of the respective sentence on page 5 line 26-27.

Comment:
*Page 7, line 10 ff: ". . .one option to decrease measurement time . . . is to increase the velocity . . .". No, this is not the case in the context of optimizing the spectral signal-to-noise-ratio (SNR). Here, only the spectral resolution and the throughput matter. I would have expected (for a given allowed integration time) to see a more pronounced reduction of error bars on the retrieved columns until a further reduction of resolution starts to decrease the contrast between the lines and the adjacent continuum. Has the spectral SNR been adjusted as function of resolution in this manner (assumption of a certain amount of available integration time)? When comparing different resolutions, one might also take into account that a larger fieldstop can be applied when resolution is reduced (increasing the signal level, favoring shorter scans even more).*
Response:
While the increase in velocity does increase the number of scans possible in the same time frame, this has no effect on the spectral S/N. The wording will be adjusted to clarify. Assuming the second comment refers to Figure 6, no, the data set has not been adjusted for the possible number of spectra within a certain time frame. The aim was to understand what impact the spectral noise has as a function of resolution on the retrieval. A figure highlighting the increase in signal-to-noise ratio (SNR) is included below in Figure 1 and will be integrated in the revised manuscript together with the following explanation: Here the increase in SNR was measured as a function of spectral resolution with a Bruker 125 HR, normalized to the SNR at $0.02$ cm$^{-1}$, i.e. a spectrum recorded with $1.0$ cm$^{-1}$ resolution has a 10 times larger SNR (see blue line). Additionally, the shorter scan length allows to record more spectra in the same time frame. Averaging leads to an increase in SNR by a $\sqrt{N}$ with N measurements (red line). The resulting black line shows the potential increase in SNR with resolution for a fixed integration time. At lower resolutions the size of the entrance aperture is limited by the size of the image of the lunar disk, rather than the resolution. Figure and discussion will be added to Section 3.4 of the final manuscript.
**Changes:**
Figure 6 was added and the paragraph page 8 line 21 - page 9 line 6 included.

Comment:
*Page 8, line 4: ". . .white noise were added.". How has this operation been performed technically? In the interferogram domain before the FFT? Note that this section does not specify (nor treats) the choice of the numerical apodization function, which seems a further important choice in addition to the scan length if reaching the best possible precision of the retrieved column is so crucial.*
Response:
Here, the interferogram was cut to obtain lower spectral resolution and then Fourier-transformed. Then the noise was added to the spectrum. In all retrievals, from solar and lunar spectra, a boxcar apodization function was applied. Note that the retrieval adjusts for the resulting sinc-shaped distortion of the spectral lines in the spectral domain. Using a different apodization function would result in information loss in the spectrum. The usage of the Boxcar apodization is mentioned in

the revised manuscript.
**Changes:**
Explanation added on page 4 line 17-18.

Comment:
*Page 12, line 10: "air-glow emissions". This study would be especially interesting if lunar spectra taken during twilight would be treated separately (spelling: airglow).*
Response:
Unfortunately, all lunar spectra recorded at a time when the sun is higher than -5° elevation have to be filtered out (see sec. 2.3). In twilight, backscatter in the atmosphere leads to a light path through the atmosphere that is not well-defined. This leads to higher retrieved DMFs, this behaviour is only partly compensated by the O2-ratio approach.
**Changes:**
Spelling corrected on page 14 line 19. No further changes to the manuscript.

Comment:
*Figure 12: Despite the fact that no biases were discovered in the September 2013 measurements, one is left with the impression that the lunar CH4 measurements in 2015 are biased high in comparison to the solar observations.*
Response:
Yes, we do not see any mechanism that would explain the apparent bias between solar and lunar xCH4 measurements in 2015 and 2016.
**Changes:**
Added brief discussion on biases on page 18 line 2 - page 19 line 6. See also the corresponding answer to review #2.

**Response to review #2 by Debra Wunch**

Comment:
*The language needs tightening - some technical concepts that are specific to TCCON or Bruker 125HR instruments that may not be familiar to the wide AMT audience are glossed over and should be written in a clearer, more general way.*
Response:
The final manuscript will be revised with emphasis on readability. We have tried to define TCCON and instrumental specific references to be more approachable by a general AMT audience.
**Changes:**
See various changes corresponding to other review comments.

Comment:
*Night time validation with aircraft or AirCore profiles would be best, but appear to be unavailable (at least, they are not mentioned in the manuscript). Perhaps this should be mentioned in the discussion or conclusions section.*
Response:
Correct, so far no aircraft campaigns above Ny-Ålesund are available. Aircore measurements are difficult. Ny-Ålesund is a coastal town surrounded by mountains and glaciers and the retrieval of the probe has to be ensured. One obvious solution to this is to deploy a guided descent, but as far as we know a secure retrieval glider is still under development. This will be mentioned in the Conclusions of the revised manuscript.
**Changes:**
Added to paragraph on page 19 line 12.

Comment:

*In Figure 14, you compare the XCH4 seasonal cycle from your lunar and solar mea- surements to the MACC model. It shows significant disagreement in summer, but not in winter, showing that the model isn't able to properly reproduce the Arctic methane seasonal cycle amplitude. Do you have any idea why? This, to me, is one of the most interesting figures/results of the paper.*

Response:

This is indeed very interesting, and something we hope to examine further. It appears that there is a general bias between the model and the solar FTS measurements with specific events in spring, where the FTS measurements show sudden decreases of the xCH4. Our current understanding is, that the model is not capable of addressing vertical transport very well. Specifically stratospheric intrusions during the breakdown of the polar vortex in spring lead to large, short-term decreases in xCH4. This is currently being investigated by using a stratospheric species as a tracer to seperate the xCH4 column in a tropospheric and stratospheric part but exceeds the scope of this paper, however we will add the above explanation to Section 5 of the final manuscript.

**Changes:**

Paragraph added to page 18 line 2 to page 19 line 6.

Comment:

*P1L4: The moon isn't a light source - it's reflected sunlight off the moon.*

Response:

Yes, in the NIR, reflected sunlight is the main component of the lunar irradiance. This will be reworded for clarity.

**Changes:**

Wording changed on page 1 line 4.

Comment:

*P1L5: I don't think you mean "parallel".*

Response:

Yes, the measurements are not actually 'parallel', but happen on consecutive days and nights. Wording has been adjusted.

**Changes:**

Wording adjusted on page 1 line 6.

Comment:

*P1L23: You don't need extended InGaAs detectors to measure above 5000 cm-1.*

Response:

Correct, reworded for clarity.

**Changes:**

Changed wavenumber range on page 4 line 2.

Comment:

*P2L18: Do you use the solar brightness fluctuation corrections for high cirrus typically employed by TCCON (embedded in I2S for DC-recorded interferograms)?*

Response:

Yes, however the effect of the correction is minimal, because in case of lunar spectra there is not enough signal with strong cirrus present. Additionally, due to the low resolution of the spectra, thin cirrus clouds typically lead to brightness fluctuations between consecutive scans and less to fluctuations within one interferogram record. This will be added to the description of the postprocessing of the spectra in Section 3.1.

**Changes:**

Explanation added to page 4 lines 19-21.

Comment:
*P2L22: 0.04 what units? mrad?*
Response:
Here: 0.04 radians. The units have been added.
**Changes:**
Units added on page 2 line 24.

Comment:
*P2L22: This sentence may be too technical for this audience. Explain that this ME and phase error are consistent with a well aligned instrument.*
Response:
The fact that these values are indicative of a well-aligned instrument has been included in the revised manuscript.
**Changes:**
Added to page 2 lines 25-26.

Comment:
*L25-35: This is too technical - please explain further.*
Response:
Assuming this comment refers to section 2.2, this will be reworded, see also the answer to the comments in review #1.
**Changes:**
Clarification added to page 3 lines 4-6.

Comment:
*P4L11: Rework sentence beginning with "Generally speaking, . . ."*
Response:
The sentence has been reworded to: Decreasing the resolution leads to a shorter measurement time and therefore allows for integration of more interferograms in the same time frame. Increasing the entrance aperture allows for more incident light on the detector which increases the signal-to-noise-ratio.
**Changes:**
Changed paragraph on page 4 lines 23-26.

Comment:
*P4L14: The entrance aperture wasn't always 3.15 mm? Please explain.*
Response:
At full moon, the entrance aperture could be set to 3.15 mm. If the moon is not full, its image on the aperture wheel requires a smaller aperture to still ensure that the aperture is uniformly lit. Additionally, the four-quadrant diode used in the tracking system, sometimes has difficulties centering the non-full lunar image, using a smaller aperture in this case, again ensures full illumination of the entrance aperture. The respective paragraph in Section 3.1 has been reworded to clarify this.
**Changes:**
Paragraph changed on page 4 lines 27-32.

Comment:
*P7L1: Please note that the large deviations are at very high SZA that would be filtered out in a typical TCCON filter. Could you make this plot for days with lower SZAs? Does it look the same?*
Response:
This has been only done for the Ny-Ålesund site, here lower SZAs are only possible in summer

and due to the midnight sun conditions, the differences between the day and night atmospheric models are smaller. However this approach can easily be adapted to other TCCON sites. It will be noted in Section 3.3 of the revised manuscript that higher SZA are generally filtered out in standard TCCON.

**Changes:**

Sentence added to page 6 lines 18-19.

Comment:

*P7L11: This worry no longer holds, given that Bruker has provided two solutions to the ghost problem (the laser sampling board potentiometer and the new M16 controllers with the XSM option), and TCCON provides a ghost removal procedure with I2S, as long as you measure simultaneously on another detector with a spectral range that is entirely within a single alias. In fact, I believe Bruker recommends 20kHz as their preferred scanner speed.*

Response:

Yes, the paragraph will be adjusted. See also the answer to review #1 regarding this issue.

**Changes:**

Sentence changed on page 8 lines 1-5. And further explanation added corresponding to answer to review #1 on page 8 lines 21 to page 9 line 6.

Comment:

*Fig 5: I see what you're trying to do with this figure, but I find it very difficult to read and interpret quantitatively. Perhaps you also need to show example slices through the 3D figures showing XCH4 vs resolution and XCH4 vs SNR.*

Response:

Figure 5 was intended to present the qualitative behaviour of different SNRs as a function of resolution. The quantitative information, e.g. xCH4 vs. resolution - for two extrem cases of SNR - is shown in Fig. 6. An additional plot will be added as described in the answer to review #1 showing the improvement of the S/N with decreasing resolution.

**Changes:**

Figure 6 and explanation on page 9 lines 1-6 have been added, as discussed in corresponding answer to review #1.

Comment:

*Fig 6: Would the x-axis scale work better as a log10 scale? Also, with the low SNR error bars as large as they are, it's difficult to see what the mean value is as a function of resolution. Perhaps you need to reduce the y-axis limits and show a representative error bar.*

Response:

The errorbars have been removed and representative errors added to the caption.

**Changes:**

Estimates added to page 9 lines 8-9 and further explanation to page 10 lines 2-6.

Comment:

*P10L10: The averaging kernel also depends on the retrieval methodology.*

Response:

Yes, this detail has been included.

**Changes:**

Included on page 11 lines 9-11.

Comment:

*P12L1: Can you assume that the total columns do not change significantly during the 24-hour period? What about drawdown from the terrestrial biosphere throughout the day and respiration at night? Is night time respiration a feature of the carbon cycle you can hope to measure with your*

*lunar measurements given the precision of your measurements? The y-axis scale is too large in Figure 9 to see whether there is any diurnal cycle in your data and models. Ditto for Figure 10.*
Response:
The standard deviation of all models are in the order of 0.2 - 0.3 ppm for xCO2 and 1.0 - 1.6 ppb for xCH4, which is an argument for the stability of the columns during the validation time period. Unfortunately the errors are too large to investigate night-time to day-time differences, e.g. due to respiration and carbon uptake, with the lunar observation presented here. The y-axis scales cannot easily be adjusted without loosing information on the lunar data points. However corresponding values for mean and standard deviation are given in Table 2.
**Changes:**
The axes in the corresponding Figure 10 and 11 have been rescaled (page 13).

Comment:
*P14L25: Remove the comma after "both".*
Response:
Done.
**Changes:**
Comma removed.

Comment:
*P14L25: The models don't capture the secular trends in XCO2 and XCH4? Why not?*
Response:
This is a misunderstanding. The models do capture the secular trends. In order to directly compare one year with another, the time series has to be detrended. This will be rephrased in the final version.
**Changes:**
Sentences on page 16 lines 20-21 have been reworded.

Comment:
*Fig 13, 14: I don't see any green dots.*
Response:
The color reference will be updated.
**Changes:**
Color reference updated in Fig. 14 & 15 and on page 16 line 35.

**A markup of the changes made to the previously submitted manuscript is attached.**

[revised manuscript text omitted]

---

## Author Response (AR2)

**Point-by-point response to editor request**

We thank the editor for the suggestions. Comment, answer and changes to the manuscript are detailed below and a markup version of the changes to the previously submitted manuscript is attached.

Editor Comment:
*At the end of Page 4 in the tracked changes version, the first reviewer's concern about potential bias due to different different spectral settings was partially answered. It is clear that the authors have chosen the correct technical setting in GGG (use of disc-averaged spectra for lunar observations rather than disc-centered spectra, as used in TCCON protocol during daytime). However, the reviewer is concerned that there could be a bias due to the choice of a different spectral settings between solar spectra and lunar (disc-averaged) spectral analysis. The first reviewer indicates this could be a source of bias, and the authors have not addressed the possibility of bias from this difference. This potential bias could be a mechanism by which the last comment of the reviewer. Please comment on on the potential for bias by this mechanism and or attempt to quantify this source of potential bias.*

Response:
Yes, the switch in retrieval procedure, i.e. using disc-averaged solar lines, could potentially introduce a bias. We have tried to address this by two means. First, the residual of the spectral fit shows no systematic deviations, that would originate in a wrong line shape of the solar lines (compare Fig. 1 below). That means, the solar lines present in the measured spectrum are well captured by the calculated solar lines. To illustrate this, we added the contribution of the solar lines in the retrieval windows to Fig. 2, showing the fit residuum. However, this only indicates, but not completely proves the absence of a bias from the solar lines. The second point is the validation with TCCON and bias discussion in Section 4. If present, a bias would be within the limits presented there.

**Changes:**
To address this, we will extend Section 3.1 as follows:
GFIT includes a setting that switches to a calculation of a disc-averaged spectrum when the moon is selected as the source. This approach leads to well captured solar lines in the spectral fit residuum (see Fig. 2) and therefore indicates the absence of a bias from using different solar line shapes, as is to be expected. A potentially introduced bias would be within the limits of the total bias to the solar measurements (e.g. $0.66 \pm 4.56$ ppm for xCO2 and $-1.94 \pm 20.63$ ppb for xCH4) as discussed in Section 4.

[revised manuscript text omitted]